# Optimal Transport-based Identity Matching for Identity-invariant Facial Expression Recognition

**Daeha Kim**
Inha University
kdhht5022@gmail.com

**Byung Cheol Song**
Inha University
bcsong@inha.ac.kr

## Abstract

Identity-invariant facial expression recognition (FER) has been one of the challenging computer vision tasks. Since conventional FER schemes do not explicitly address the inter-identity variation of facial expressions, their neural network models still operate depending on facial identity. This paper proposes to quantify the inter-identity variation by utilizing pairs of similar expressions explored through a specific matching process. We formulate the identity matching process as an Optimal Transport (OT) problem. Specifically, to find pairs of similar expressions from different identities, we define the inter-feature similarity as a transportation cost. Then, optimal identity matching to find the optimal flow with minimum transportation cost is performed by Sinkhorn-Knopp iteration. The proposed matching method is not only easy to plug in to other models, but also requires only acceptable computational overhead. Extensive simulations prove that the proposed FER method improves the PCC/CCC performance by up to 10% or more compared to the runner-up on wild datasets. The source code and software demo are available at https://github.com/kdhht2334/ELIM_FER.

## 1 Introduction

For sophisticated human-computer interaction (HCI), facial expression recognition (FER) must be able to accurately predict the emotions of people with different personalities or expressive styles. On the other hand, to express the type and intensity of emotions in detail, VA FER, which annotates facial expressions in a continuous domain based on two axes of Valence (V) and Arousal (A), is attracting attention [36]. Here, valence and arousal indicate the positive/negative of emotions and the degree of activation, respectively. Recently, many techniques for analyzing complex or fine-grained expressions have been developed to improve the regression or prediction performance of VA FER [16, 27, 17]. However, it is still challenging to properly learn facial expressions that differ from person to person due to heterogeneity in identity (ID) attributes such as age, gender, and culture.

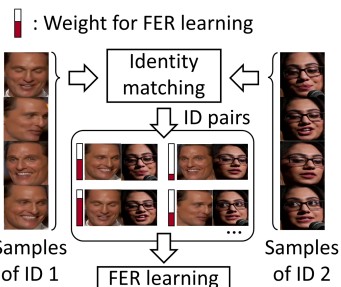

Figure 1: Conceptual illustration of ELIM. ELIM utilizes identity pairs that are advantageous for FER learning.

Thus, so-called ID-aware FER is receiving a lot of attention nowadays. For example, [22, 2] learned emotional factors inherent in strong facial expression samples through adversarial learning, and generated ID samples favorable to FER through GAN [14]. However, it is observed that conventional models have difficulty learning the unique properties of facial expressions inherent in each ID, and above all, they are still dependent on the ID change.

36th Conference on Neural Information Processing Systems (NeurIPS 2022).

We believe that the inter-ID variation of facial expressions is the key to solving this problem. To analyze the inter-ID variation, we propose a novel FER learning based on pair-wise similarity by finding pairs of similar expressions from different IDs. This idea was inspired by cognitive research [6, 48, 18] that analyzes facial expressions through comparison between IDs. This paper realizes our idea through a novel framework named Expression Learning with Identity Matching (ELIM). Specifically, ELIM finds ID pairs with similar expressions through ID matching based on the optimal transport (OT) problem [8], and assigns relevance weights according to the similarity of ID pairs (see Fig. 1). Next, based on the relevance weights, we compute and compare the tendency of differences in facial expressions between IDs, i.e., the ID shift of facial expressions. As discussed in Sec. 4, ID-invariant representations are generated by ID shift vectors quantified from ID pairs. Finally, FER learning is performed through risk minimization [43].

Contributions of this paper are summarized as follows: **(I)** To realize ID-invariant FER, we propose an FER framework ELIM that adaptively considers inter-ID variation of facial expressions. Specifically, ELIM learns facial expressions based on the similarity (or dissimilarity) between IDs obtained through ID matching (see Fig. 4). **(II)** The state-of-the-art (SOTA) performance of ELIM is verified through extensive experiments on various real-world datasets. Especially, ELIM works more accurately compared to prior arts even for samples in which inconsistency between facial expressions and the emotion label predictions exists (see Fig. 3).

## 2  Related Work

**Valence-Arousal (VA) FER** is an approach based on the circumplex model [36] in which emotions are distributed in a two-dimensional (2D) circular space. VA FER has a wide research spectrum, from formulating mapping between features and continuous VA labels [17] to mapping complex and high-dimensional facial expression information on VA space [27, 37]. Recently, some interesting studies have been reported on the so-called ID-invariant FER, which is designed to be robust to even unseen IDs. For example, [2] introduced a generative model to generate and analyze similar expression examples from different IDs. However, [2] is fundamentally dependent on the performance of the generative model, and its performance is limited because only a single pair is compared to analyze the expression difference between IDs. From another point of view, several FER methods tried to learn intra-identity discrepancy of samples with different facial expressions [47, 39]. However, since they do not explicitly consider inter-ID variations, they inevitably suffer from performance degradation for IDs that was never learned [41]. [5] proposed a personalized affective module based on an adversarial autoencoder to learn features and generate samples having different expressions. However, like [47, 39], this VA FER method also shows limited performance for unseen target ID. On the other hand, since ELIM reflects the relative traits of expression between IDs in model learning, it can achieve reliable performance even on the test dataset including unseen IDs.

**Distribution shift** has been treated as very important not only in FER but also in various machine learning tasks. For example, distribution shifts between set of source domains and unseen target domains [32, 49] are seriously considered in the field of out-of-domain (OoD) generalization [34, 15]. In particular, normalization methods [52, 12] that generate domain-invariant representations by quantifying distribution shifts through domain-specific statistic are being actively studied in the OoD generalization field. ELIM is differentiated from conventional normalization methods [52, 12] in that it defines statistics from the OT problem that automatically matches pairs of similar facial expressions. Note that ELIM is the first case applying the distribution shift concept to the field of VA FER (cf. Problem 1). Specifically, ELIM regards ID as a domain and quantifies ID shift vectors. By normalizing features with the corresponding ID vectors, ID-invariant representations can be generated.

## 3  Preliminaries

This section describes the problem formulation of ELIM and briefly reviews the OT problem responsible for optimal resource allocation.

### 3.1  Problem Formulation with Nomenclature

This paper focuses on the regression task to estimate VA labels $\boldsymbol{y}(\in \mathbb{R}^2) \subset \mathcal{Y}$ from features $\boldsymbol{z}(\in \mathbb{R}^d) \subset \mathcal{Z}$. Here, $\mathcal{Y}$ and $\mathcal{Z}$ stand for the label and feature spaces, respectively. The VA FER task

generally assumes that labels depend on facial expressions rather than IDs. However, this assumption does not always hold due to ID shift, i.e., a phenomenon in which (marginal) distributions of IDs are shifted by their attributes even though the label prediction mechanism is the same: $p^1(\boldsymbol{y}|\boldsymbol{z}) = p^2(\boldsymbol{y}|\boldsymbol{z})$ and $p^1(\boldsymbol{z}) \neq p^2(\boldsymbol{z})$. Let $p^1$ and $p^2$ denote distributions of two different IDs. We aim to find a predictor $f : \mathcal{Z} \to \mathcal{Y}$ that works robustly even when label predictions are inconsistent with the corresponding facial expressions. Formally,

**Problem 1.** *Let $Z^e = \{\boldsymbol{z}^e\}_{e \in \mathrm{supp}(\mathcal{E}_{tr})}$ be the training-purpose feature set collected from multiple IDs. The corresponding VA label set $Y^e$ is defined similarly. $\mathcal{E}_{tr}$ is an index set of training environments and $\mathrm{supp}(\cdot)$ stands for its support set. The cardinality of $\mathcal{E}_{tr}$ is $N$. Then, the risk of predictor $f$ for ID $e$ is formulated as follows:*

$$\arg\min_f \max_{e \in \mathrm{supp}(\mathcal{E}_{tr})} \mathbb{E}^e[\mathcal{L}(f(Z^e), Y^e)] \tag{1}$$

*where $\mathcal{L}$ is a loss function based on mean-squared error (MSE).*

Note that the goal of Problem 1 is to minimize the empirical risk [43] of $f$ from as many IDs as possible. To minimize the risk of Eq. 1 derived to generalize ID shifts, we design a model to learn representations of facial expressions invariant to ID shifts (cf. Sec. 4.2).

## 3.2 Review of Optimal Transport

This section briefly review the OT problem that is being applied to various real-world applications such as speech signal processing and biomedical technology [30, 46]. In this paper, the OT problem is applied to the VA FER task for a similar purpose to the above studies. OT is known as a tool for measuring the optimal distance between two distribution sets. For example, suppose you are transporting a product from a set of suppliers $\boldsymbol{a} = \{a_i | i = 1, 2, \cdots, n\}$ to a set of demanders $\boldsymbol{b} = \{b_j | j = 1, 2, \cdots, m\}$. Also, assume that the $i$-th supplier $a_i$ has some unit(s) of products, and the $j$-th demander $b_j$ requires some unit(s) of products. Here, $a_i$ and $b_j$ mean the number of product(s) in a double sense. The cost per unit transported from supplier $i$ to demander $j$ is $C_{i,j} \in \mathbb{R}_+^{n \times m}$, and the number of units transported is $X_{i,j} \in \mathbb{R}_+^{n \times m}$. The objective of the OT problem is to search the cheapest flow $X^* = \{X_{i,j} | i = 1, \cdots, n, j = 1, \cdots, m\}$ that can transport all products from suppliers to demanders at the minimum cost:

$$\mathrm{OT}(\boldsymbol{a}, \boldsymbol{b}) = \min_X \sum_{i,j} C_{i,j} X_{i,j},$$
$$\text{s.t. } \sum_{i=1}^n X_{i,j} = b_j, \ \sum_{j=1}^m X_{i,j} = a_i, \ \sum_{i=1}^n a_i = \sum_{j=1}^m b_j. \tag{2}$$

In order to find the solution $X^*$ of the above problem, we can employ linear programming as a tool. However, since this tool does not necessarily provide a unique solution and it requires near-cubic complexity [1], it is difficult to apply to high-dimensional features. So, we address this issue through a fast iteration solution, i.e., Sinkhorn-Knopp [8] (cf. **Appendix A**). Sinkhorn-Knopp method based on the matrix scaling algorithm can obtain the unique optimal solution $X^*$ with near-square complexity [3]. Since this method computes $X^*$ by aggregating the importance between all features, it can analyze both global and local distributions of data. In this paper, we define $a_i$ and $b_j$ from the features of each ID to capture the global representation of facial expressions, and perform the ID matching process through the Sinkhorn-Knopp method.

## 4 Proposed Method: ELIM

This section proposes Expression Learning with Identity Matching (ELIM) for VA FER. ELIM consists of Sinkhorn-Knopp-based ID matching and ID-invariant expression learning.

**Pre-processing.** Since ELIM regards ID as a domain, samples in the mini-batch need to be grouped according to ID. One reference ID is randomly selected among $N$ IDs in the mini-batch and the remaining $N-1$ IDs are our target. Then, a feature set $Z^e$ is constructed by the backbone ($f_\phi$) [29, 19, 40] on an ID basis. Next, ID matching process is performed between each target ID and the reference ID, thereby ID pairs corresponding to the target ID are constructed.

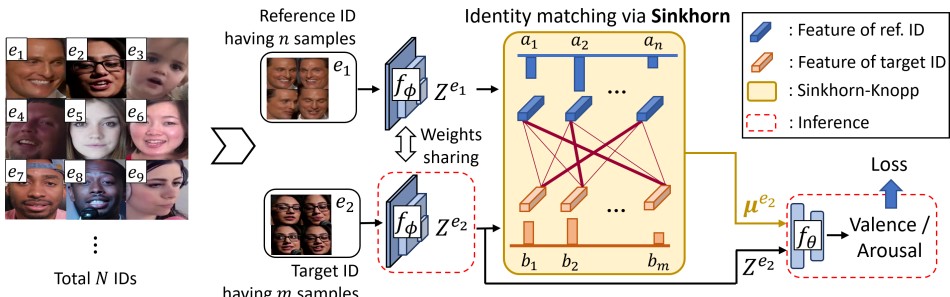

Figure 2: Overview of the ELIM. Among the total $N$ IDs in the mini-batch, a reference ID, i.e., $e_1$ is randomly chosen. The rest become target IDs $e_{2,\cdots,N}$. For convenience, only one target ID $e_2$ is considered here. First, $f_\phi$ encodes feature sets. The ID matching cost is defined from the feature sets, and then node weights $a_i$ and $b_j$ are computed. Next, optimal matching flows having the minimum matching cost are generated by Sinkhorn-Knopp iteration. Finally, the ID shift vector $\boldsymbol{\mu}^{e_2}$ corresponding to the target ID is derived. $f_\theta$ is learned to generate ID-invariant representations by receiving $Z^{e_2}$ and $\boldsymbol{\mu}^{e_2}$ as inputs. During inference, only $f_\phi$ and $f_\theta$ are used without additional cost.

## 4.1 Identity Matching via Sinkhorn-Knopp Iteration

Early ID matching approach focused on calculating differences between features based on distance metrics such as L2-norm and contrastive loss [20, 22]. However, since conventional methods based on inter-sample matching can quantify even differences in attributes (e.g., hair, skin, etc.) unrelated to facial expressions, they cannot be free from the influence of ID. In other words, previous works are not robust to expression differences between IDs, i.e., inter-ID variation. To overcome this drawback, even inter-ID variation should be learned. To derive the information about inter-ID variation, reliable ID pairs are required. To acquire reliable ID pairs, we match samples from different IDs by using Sinkhorn-Knopp iteration [8] which can consider both global and local distributions of samples.

As in Fig. 2, this section uses only a reference ID $e_1$ and a target ID $e_2$ as an example for convenience. Let their feature sets be $Z^{e_1} \in \mathbb{R}^{d \times n}$ and $Z^{e_2} \in \mathbb{R}^{d \times m}$, respectively. Here, the cardinality of $Z^{e_1}$ and $Z^{e_2}$ is $n$ and $m$, respectively. First, matching costs and relevance weights are generated from $Z^{e_1}$ and $Z^{e_2}$. The cost matrix $C$ is defined based on cosine similarity so that a pair of features with a similar expression has a smaller matching cost: $C_{i,j} = 1 - \frac{(\boldsymbol{z}_i^{e_1})^T \boldsymbol{z}_j^{e_2}}{\|\boldsymbol{z}_i^{e_1}\| \|\boldsymbol{z}_j^{e_2}\|}$. Here, $\|\cdot\|$ indicates L2-norm. Then, relevance weights $a_i$ and $b_j$ are computed to capture the global representation inherent in each ID (cf. Sec. 4.3). Using the relevance weights, the Sinkhorn-Knopp iteration computes the optimal flow matrix $X^*$ (cf. **Appendix A**). As a result, based on $C$ and $X^*$, the ID shift of the target ID is calculated as follows:

$$\mu_j^{e_2} = \sum_{i=1}^{n} C_{i,j} X_{i,j}^*, \quad j = 1, 2, \cdots, m. \tag{3}$$

$\mu_j^{e_2}$ is a weighted cosine distance with the optimal transportation plan $X^*$ between the $j$-th target sample and the reference samples as a weight. ID shift vectors $\boldsymbol{\mu}^{e_2}, \cdots, \boldsymbol{\mu}^{e_N}$ of all target IDs computed in the same way are used for feature normalization process of the next section.

## 4.2 Identity-Invariant Expression Learning

This section describes the process of generating and learning the ID-invariant representation of the target feature set $Z^e$ using the ID shift vector $\boldsymbol{\mu}^e$. First, inspired by several normalization techniques [52, 12] that perform domain-specific adaptation or generalization of features, we define the ID-invariant representation by using $\boldsymbol{\mu}^e$ and $\boldsymbol{\sigma}^e$ as shifting and scaling parameters, respectively:

$$\hat{Z}^e = \frac{Z^e - \boldsymbol{\mu}^e}{\boldsymbol{\sigma}^e + \epsilon}, \qquad \forall e \in \{e_2, \cdots, e_N\}, \tag{4}$$

where $\sigma_j^e = \sqrt{\frac{\sum_{i=1}^{d}(Z_{i,j}^e - \mu_j^e)^2}{d}}, j = 1, \cdots, m$. Since $\boldsymbol{\mu}^e$ and $\boldsymbol{\sigma}^e$ captured from a specific ID are used, the normalization of Eq. 4 can remove ID-specific information from the model. Next, the normalized

$d$-dimensional feature set $\hat{Z}^e$ is mapped onto the 2D VA space:

$$\hat{Y}^e = f_\theta(\hat{Z}^e) = \boldsymbol{\gamma}^T \hat{Z}^e + \boldsymbol{\beta}, \qquad \forall e \in \{e_2, \cdots, e_N\}, \tag{5}$$

where $\theta = \{\boldsymbol{\gamma} \in \mathbb{R}^{d \times 2}, \boldsymbol{\beta} \in \mathbb{R}^2\}$. Affine parameter $\theta$ is trained to adapt to the normalized features of target IDs in $f_\theta$ of Eq. 5, which behaves similarly to adaptive instance normalization [21]. That is, a given model is encouraged to learn an ID-invariant representation [31, 45, 12]. Finally, the risk loss [43] between $\hat{Y}^e$ and the VA label set $Y^e$ is calculated:

$$\mathcal{L} := \min_{\phi, \theta} \sum_{e \in \mathrm{supp}(\mathcal{E}_{tr})} \mathbb{E}_{(Z,Y) \sim p^e} \left( f_\theta(\hat{Z}^e) - Y^e \right)^2 \tag{6}$$

Note that Eq. 6, which regresses predictions and labels in terms of MSE, calculates empirical risk loss through Monte-Carlo sampling.

---

**Algorithm 1** Training Procedure of ELIM

---

**Require:** # IDs $N$, learning rate $lr$, initialize $\mathcal{L}_{main}$ to zero, initialize parameters $(\phi, \theta)$ to Normal
 1: **while** *not* converge $(\phi, \theta)$ **do**
 2:     $Z^e \leftarrow$ ID grouping and feature encoding via $f_\phi$, $\forall e \in \{e_1, \cdots, e_N\}$
 3:     **for** $i = 2, \cdots, N$ **do**
 4:         $\boldsymbol{\mu}^{e_i} \leftarrow$ ID matching and ID shift vector calculation (Eq. 3)
 5:         $\hat{Z}^{e_i}, \hat{Y}^{e_i} \leftarrow$ Normalization and mapping into VA space (Eqs. 4 and 5)
 6:         $\mathcal{L}_{main} := \mathcal{L}_{main} + \mathrm{MSE}(\hat{Y}^{e_i}, Y^{e_i}) \leftarrow$ Update the main loss (Eq. 6)
 7:     **end for**
 8:     $\mathcal{L}_{total} = \mathcal{L}_{main} + \mathcal{L}_{va} \leftarrow$ Calculate the total loss
 9:     $(\phi, \theta) := (\phi, \theta) - lr\nabla_{(\phi, \theta)}\mathcal{L}_{total} \leftarrow$ Update learnable model parameters $(\phi, \theta)$
10: **end while**

---

**Training procedure.** Algorithm 1 describes the details of the losses of ELIM calculated every step. Reference feature set $Z^{e_1}$ and target feature sets $Z^{e_2, \cdots, N}$ quantify ID shift vectors $\boldsymbol{\mu}^{e_2, \cdots, N}$ through ID matching. ID-invariant representations are generated from these vectors and $Z^{e_2, \cdots, N}$, and then the empirical risk loss $\mathcal{L}_{main}$ is calculated in the VA space. With $\mathcal{L}_{va}$ [50] calculated by MSE between unnormalized feature set $Z^e$ and label $Y^e$, parameters $(\phi, \theta)$ are updated until convergence.

## 4.3 Relevance Weight Design

Since the matching flow $X$ is determined based on relevance weights, i.e., $\boldsymbol{a}$ and $\boldsymbol{b}$, the design of the relevance weights is important for effective ID matching. This section describes the design process with the aforementioned $Z^{e_1} \in \mathbb{R}^{d \times n}$ and $Z^{e_2} \in \mathbb{R}^{d \times m}$ as examples.

**Sampling reliable features.** If a reference ID has a weak facial expression, the difference in facial expression with other IDs will not be noticeable, which indicates the difficulty in ID matching. In other words, the feature of such a reference ID, i.e., $\boldsymbol{z}^{e_1}$ is not suitable for matching. Therefore, we set the norm $\|\boldsymbol{y}^{e_1}\|$ of the label corresponding to $\boldsymbol{z}^{e_1}$ to the sampling score $\pi$ of $\boldsymbol{z}^{e_1}$: $\pi = \|\boldsymbol{y}^{e_1}\|/(\sum_i \|\boldsymbol{y}_i^{e_1}\|)$, $i = 1, \cdots, n$. $\boldsymbol{z}^{e_1}$s corresponding to top $k$ elements in terms of $\pi$ are sampled. However, since the sampling of $\boldsymbol{z}^{e_1}$ is non-differentiable, we employ a re-parameterized gradient estimator with the Gumbel-topk [26]:

$$g = \log \pi + \mathrm{Gumbel}(0, 1) \tag{7}$$

Retrieving the $k(\leq n)$ largest values $\mathrm{topk}(g_1, \cdots, g_n)$ will give us $k$ samples *without replacement*. Finally, $Z^{e_1}, Z^{e_2} \in \mathbb{R}^{d \times k}$s retrieved in the same way are used to calculate relevance weights.

**Computing relevance weights.** If weights are generated only from features of either reference ID or target ID, it will be difficult to capture the global representation of facial expression. Instead, if aggregated information of features between IDs is used for weight calculation, it will become easier to find ID pairs that are advantageous for FER. This can be proven experimentally (cf. Table 4). Inspired by the feature representation of a few-shot classification task [38], we use the average of $\boldsymbol{z}^{e_2}$s as support of $\boldsymbol{z}^{e_1}$ to calculate the reference ID's relevance weight $\boldsymbol{a}$:

$$a_i = \frac{\hat{a}_i}{\|\hat{\boldsymbol{a}}\|}, \quad \text{s.t. } \hat{a}_i = \left[ \boldsymbol{z}_i^{e_1} \cdot \sum_{j=1}^m \frac{\boldsymbol{z}_j^{e_2}}{m} \right]_+ \quad i = 1, 2, \cdots, n, \tag{8}$$

where $[\cdot]_+$ indicates the clipping of negative value(s). Another relevance weight $\boldsymbol{b}$ of the target ID is calculated similarly. The effect of the proposed relevance weight will be analyzed in Sec. 5.4.

## 5 Experiments

### 5.1 Datasets and Implementation Details

**Datasets.** We adopted public databases only for research purposes, and informed consent was obtained if necessary. **AFEW-VA** [28] derived from the AFEW dataset [10] consists of about 600 short video clips annotated with VA labels frame by frame. Evaluation was performed through cross validation at a ratio of 5:1. **Aff-wild** [50] is the first large-scale in-the-wild VA dataset that collects the reactions of people who watch movies or TV shows. Aff-wild with 298 videos amounts to over 30 hours in length, whose number of frames is about 1.18M. Since the test data of Aff-wild was not disclosed, this paper adopted the sampled train set for evaluation purpose in the same way as previous works [17, 22]. **Aff-wild2**, which added videos of spontaneous facial behaviors to Aff-wild, was also released [25]. This database consists of about 1.41M frames. Since subjects of different ethnicity and age groups were newly added, Aff-wild2 can further enhance the reliability of FER performance.

**Configurations.** All models were implemented in PyTorch, and the following experiments were performed on Intel Xeon CPU and NVIDIA RTX A6000 GPU. Each experiment was repeated five times. The light-weighted AlexNet (AL-tuned) [29], ResNet18 (R18) [19], and Mlp-Mixer (MMx) [40] were chosen as $f_\phi$, respectively. The mini-batch size for those models was set to 512, 256, and 128, respectively (cf. **Appendix B** for model details). Learnable parameters $(\phi, \theta)$ were optimized with Adam optimizer [23]. The learning rate (LR) of $\phi$ and $\theta$ was set to 5e-5 and 1e-5, respectively. LR decreases 0.8-fold at the initial 5K iterations and 0.8-fold every 20K iterations. $\epsilon$ of Eq. 4 is 1e-6. At each iteration, the number of ID, i.e., $N$ was set to 10 by default. The size $d$ of $\boldsymbol{z}$ is 64, and the size $k$ of Gumbel-based sampling in Sec. 4.3 is 10. Note that the cardinality of $Z^e$ is always greater than or equal to $k$. For face detection, deep face detector [51] was used, and the detected facial regions were resized to 224×224 through random cropping (center cropping when evaluating). ID grouping is performed based on folders in the database without separate clustering tools. Inference is performed through backbone ($f_\phi$) and projector ($f_\theta$) without Eq. 3 and Eq. 4.

**Evaluation metrics.** Root mean-squared error (RMSE) and sign agreement (SAGR) were used to measure the point-wise difference and overall degree of expressions. In addition, Pearson correlation coefficient (PCC) and concordance correlation coefficient (CCC) were employed to measure the expression tendency (cf. **Appendix C** for metric details). Since the objective of FER learning is to simultaneously achieve the minimization of RMSE and the maximization of PCC/CCC [22, 27], $\mathcal{L}_{PCC}$ and $\mathcal{L}_{CCC}$ are used together to learn $\mathcal{L}_{va}$, where $\mathcal{L}_{C(/P)CC} = 1 - \frac{C(/P)CC_v + C(/P)CC_a}{2}$.

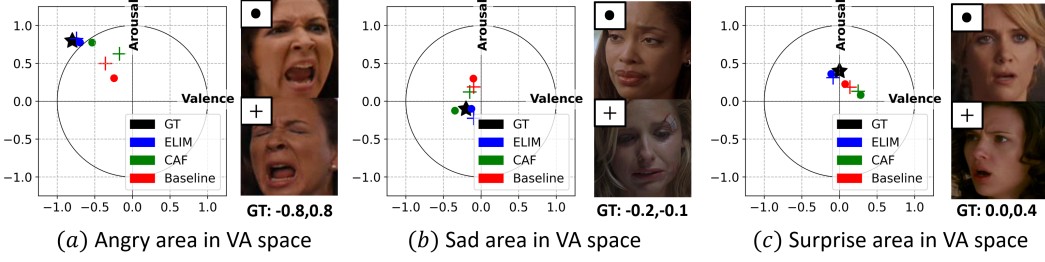

(*a*) Angry area in VA space     (*b*) Sad area in VA space     (*c*) Surprise area in VA space

Figure 3: Verification experiments of ID shift when $p^\bullet(\boldsymbol{y}|\boldsymbol{z}) = p^+(\boldsymbol{y}|\boldsymbol{z})$ but $p^\bullet(\boldsymbol{z}) \neq p^+(\boldsymbol{z})$. Predictions of the two images are located in VA space as $\bullet$ and $+$, respectively. Best viewed in color.

### 5.2 Idea Verification of ELIM

To explain the mechanism of ELIM more clearly, we present experimental answers to the most curious questions: **Q1.** Is ELIM robust when there is a discrepancy between facial expression and its label prediction? **Q2.** Is ELIM successful in learning identity (ID)-invariant features? For a fair experiment, we compared the proposed method with CAF [22] of SOTA performance as well as the baseline

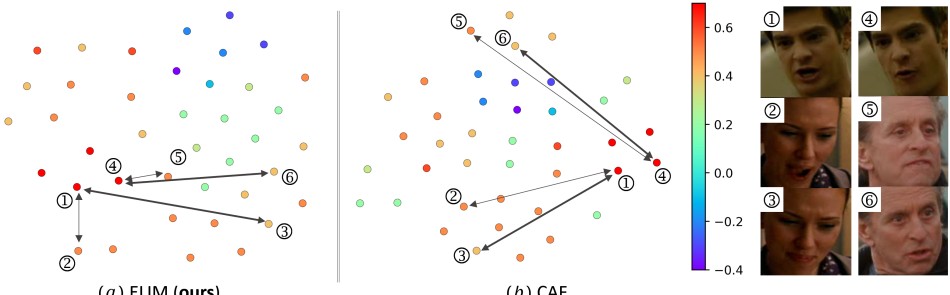

Figure 4: Verification of ID-invariant representation ability between ELIM and CAF. To effectively examine the ID dependency, we compared the distances between an anchor expression sample (e.g., ①) and two different expression samples of the same ID (e.g., ② and ③). Here, the color bar and black lines indicate the range of arousal values and arousal differences, respectively. The larger the difference, the thicker the line. Ten IDs randomly sampled from the test set were used for this experiment.

composed of only CNN and FC layers [29]. All methods were trained with AL-tuned backbone on the AFEW-VA dataset under the same experimental condition as ELIM.

**A1. Analysis of inconsistency cases.** The main goal of ELIM is to accomplish robust performance even in cases where expressions and corresponding label predictions are inconsistent. To verify this, we implement an ID shift case (cf. Sec. 3.1). In detail, we examine how robust each method is to ID shifts for facial images with the same GT but different IDs. Fig. 3 illustrates the prediction results in the VA space. Compared with CAF and baseline, ELIM not only estimates GT precisely, but also outputs relatively consistent prediction results. For more examples, refer to **Appendix D**.

**A2. Visualization of ID-invariant representation ability.** We visualized feature distributions of ELIM and CAF through t-SNE [42]. Fig. 4 shows the result of comparing the distance between an anchor expression sample and two different expression samples with the same ID. Note that in the case of ELIM, samples with different IDs but similar expressions are located closer in the feature space (see ①-② and ④-⑤ in Fig. 4(a)). Meanwhile, in CAF, samples with the same ID tend to be located close to each other (see ②-③ and ⑤-⑥ in Fig. 4(b)). ELIM adaptively considers inter-ID variations of facial expressions through the OT problem, but CAF seems to have an ID-dependent feature distribution because it has no means to explicitly consider them. For visualization results and qualitative experiments of other feature distributions, refer to **Appendix D**.

Table 1: Experimental results on Aff-wild dataset. * was evaluated on Aff-wild's *test* set using ResNet50 backbone. 'tuned' refers to the parameter lightweight version.

| Methods | Backbone | # of params | RMSE (↓) | | PCC (↑) | | CCC (↑) | |
|---|---|---|---|---|---|---|---|---|
| | | | (V) | (A) | (V) | (A) | (V) | (A) |
| Hasani *et al.* [16] | Inception-ResNet | ∼56M | 0.27 | 0.36 | 0.44 | 0.26 | 0.36 | 0.19 |
| BreG-NeXt [17] | ResNeXt50 | 3.1M | 0.26 | 0.31 | 0.42 | 0.40 | 0.37 | 0.31 |
| MIMAMO [9]* | ResNet50+GRU | >25.6M | - | - | - | - | 0.58 | 0.52 |
| CAF [22] | AL-tuned | 2.7M | 0.24 | 0.21 | 0.55 | 0.57 | 0.54 | 0.56 |
| CAF [22] | ResNet18 | 11M | 0.22 | 0.20 | 0.57 | 0.57 | 0.55 | 0.56 |
| ELIM (ours) | AL-tuned | 2.6M | 0.228 | 0.199 | 0.634 | 0.638 | **0.628** | 0.611 |
| | ResNet18 | 11.7M | 0.217 | **0.197** | 0.635 | 0.602 | 0.609 | 0.570 |
| | Mlp-Mixer | 59.1M | **0.214** | **0.197** | **0.645** | **0.684** | 0.611 | **0.631** |

## 5.3 Comparison with State-of-the-art Methods

Table 1 compares ELIM and several existing methods in terms of various evaluation metrics for the Aff-wild dataset. Note that ELIM shows better CCC performance while having a smaller parameter size than other methods including CAF. For example, ELIM with AL-tuned has about 21 times smaller

parameter size than Hasani *et al.* [16] but improves CCC(V) by about 26%. Furthermore, ELIM shows 8.8% higher CCC(V) than a prior art CAF [22]. In addition, an experiment was conducted using the recently popular Transformer Mlp-Mixer [40] as a backbone. Although the parameter size increased, a noticeable performance improvement was observed. ELIM with MMx showed about 11% better PCC(A) than CAF, and its RMSE and CCC were also improved significantly.

Table 2 shows the spontaneous expression recognition performance of short video clips through a total of four evaluation metrics. Note that ELIM, which achieved SOTA performance, and CAF, which took 2nd place, are both FER methods designed to be robust against changes in identity's expression. As a result, this suggests that it is more meaningful to extract ID-invariant features than to focus only on changes in the expression itself. And, ELIM showed better performance than CAF in terms of CCC metric, which is important for expression tendency analysis. For example, ELIM with R18 showed CCC higher than CAF with R18 by about 2% and 7% in terms of Valence and Arousal, respectively. Considering that the Arousal axis indicating the degree of activation has higher prediction difficulty than the Valence axis indicating positive and negative emotions, such a performance improvement is sufficiently significant. In addition, even in the most difficult Aff-wild2, ELIM showed better CCC (mean) than the latest FER method AP [37] (cf. Table 3). These results suggest that further performance improvement is expected if temporal information is applied to static image-based ID-invariant methods.

Table 2: Experimental results on AFEW-VA dataset.

| Case | Methods | RMSE (↓) | | SAGR (↑) | | PCC (↑) | | CCC (↑) | |
|---|---|---|---|---|---|---|---|---|---|
| | | (V) | (A) | (V) | (A) | (V) | (A) | (V) | (A) |
| Static | Mitenkova *et al.* [33] | 0.40 | 0.41 | - | - | 0.33 | 0.42 | 0.33 | 0.40 |
| | Kossaifi *et al.* [27] | 0.24 | 0.24 | 0.64 | 0.77 | 0.55 | 0.57 | 0.55 | 0.52 |
| | CAF (AL-tuned) [22] | 0.20 | 0.20 | 0.66 | 0.83 | 0.67 | 0.63 | 0.58 | 0.57 |
| | CAF (R18) [22] | 0.17 | 0.18 | 0.68 | 0.87 | 0.67 | 0.60 | 0.59 | 0.54 |
| | ELIM (AL-tuned) | 0.186 | 0.198 | 0.692 | 0.815 | 0.680 | 0.645 | 0.602 | 0.581 |
| | ELIM (R18) | 0.168 | 0.170 | 0.723 | 0.876 | 0.651 | 0.679 | 0.615 | **0.614** |
| | ELIM (MMx) | **0.167** | **0.164** | **0.747** | **0.877** | **0.704** | **0.707** | **0.643** | 0.598 |
| Temporal | Kollias *et al.* [24] | - | - | - | - | 0.51 | 0.58 | 0.52 | 0.56 |
| | HO-Conv [27] | 0.28 | 0.19 | 0.53 | 0.75 | 0.12 | 0.23 | 0.11 | 0.15 |
| | HO-Conv [27]-trans. | 0.20 | 0.21 | 0.67 | 0.79 | 0.64 | 0.62 | 0.57 | 0.56 |

Table 3: Results on the *validation* set of Aff-wild2.

| Methods | CCC (V) | CCC (A) | Mean |
|---|---|---|---|
| ConvGRU [7] | 0.398 | 0.503 | 0.450 |
| Self-Attention [44] | 0.419 | 0.505 | 0.462 |
| AP [37] | 0.438 | 0.498 | 0.468 |
| ELIM (AL) | 0.451 | 0.478 | 0.465 |
| ELIM (R18) | 0.449 | 0.496 | 0.473 |
| ELIM (MMx) | 0.498 | 0.493 | 0.495 |

Table 4: Effectiveness of OT problem on different settings.

| | Weighting methods | Sampling methods | Elapsed time (sec.) | CCC (mean) |
|---|---|---|---|---|
| Sinkhorn | Self-side Relevance | None | 1.97 | 0.580 |
| | | | 2.44 | 0.587 |
| | Self-side Relevance | Gumbel | 0.26 | 0.583 |
| | | | 0.28 | 0.592 |
| QPTH | Self-side Relevance | Gumbel | 0.97 | 0.586 |
| | | | 1.05 | 0.589 |

## 5.4 Ablation Studies and Discussions

Our ablation studies aim to answer: **Q3.** Is relevance weight effective? **Q4.** Is ID matching valid? **Q5.** Is the number of training IDs the larger the better? **Q6.** What other factors can be analyzed for ID shift? All experiments in this section were performed under the same setting as AL-tuned of Table 2.

**A3. Effectiveness analysis on relevance weights.** Sinkhorn-Knopp [8] depends on the size and number of relevance weights. Table 4 shows the effectiveness of this fast iteration tool in three settings. First, let's examine the effect of the optimization way on the OT problem. As a benchmark group of Sinkhorn-Knopp, we chose QPTH [4] that is a representative optimization tool based on the interior point method of factorized KKT matrix. Looking at the 3rd and 5th rows of Table 4, Sinkhorn

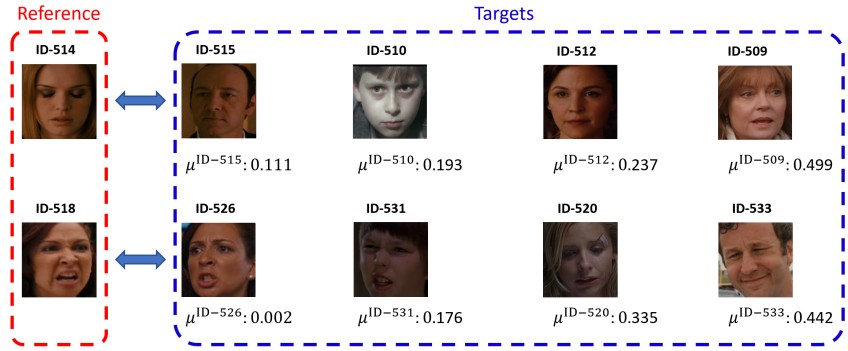

Figure 5: Verification of ID matching on the validation split of AFEW-VA.

provides approximately 3.7 times faster operation speed than QPTH while showing almost equivalent CCC performance. This proves that Sinkhorn, which has been often used to solve transportation problems of high-dimensional features, has the advantage of fast operation speed because it is mainly based on iterative updates of vectors. Next, note that the relevance weight according to Eq. 8 shows better performance improvement than the self-side (non-relevance) weight despite a slight increase in computation. Here, 'self-side' indicates determining the weights with the L2-norm of the individual features of each ID. Looking at the 3rd and 4th rows of Table 4, the relevance weight provides a significant CCC performance improvement of about 1%. This suggests that using relevance weights is valuable enough. Finally, Gumbel-based sampling has a processing speed of about 8.7 times faster than non-sampling, while providing higher CCC (mean) by 0.5% (see 2nd and 4th rows). Therefore, we can argue that when calculating ID shifts, it is very important to select samples showing clear differences in facial expressions as in ELIM. See **Appendix D** for additional results from Sinkhorn.

**A4. Validation of ID matching.** When training our model, we adopted a sufficiently large sized mini-batch. As a result, we did not observe any extreme case where the facial expressions of all samples between IDs are totally different each other. If the valence signs of facial expressions between ID groups are all different, ID-dependent features can be generated through Eq. 4 due to the matching between samples of different facial expressions. However, as mentioned above, this phenomenon occurs very rarely. Also, since ELIM explores a solution through iterative training, such a case has little effect on the overall performance of our model. Fig. 5 shows the ID shift value $(\mu^{\text{ID}-\#})$ between the reference and target samples. For example, when ID-518 is set to a reference, a sample with similar expression such as ID-526 has a small ID shift value, whereas a sample with opposite valence sign (i.e., ID-533) has a relatively large ID shift value. For convenience, only examples of one sample per ID are shown in this figure.

**A5. Influence of the number of training IDs.** ELIM, which performs matching of ID pairs through the OT problem, assumes a sufficient number of IDs. To analyze the effect of the number of training IDs, we trained ELIM with or without $\mathcal{L}_{va}$. Fig. 6 shows that the performance of the two models increases at almost the same rate as the number of training IDs increases. This prove that a sufficient number of training IDs is required, and that ELIM with more training IDs has the potential to bring about a higher performance improvement.

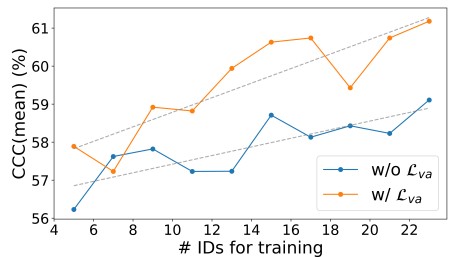

Figure 6: Influence of the number of training ID. In case of 'w/o $\mathcal{L}_{va}$', only $\mathcal{L}_{main}$ and $\mathcal{L}_{C(/P)CC}$ were used for model training.

**A6. Other factors that can influence FER learning.** Since inter-ID variation can be sufficiently shifted by change of age or gender [13, 11], age, gender, race, etc. can also be used as indicators to measure inter-ID variation. As a fragmentary example, there is a technical report that analyzes the tendency of differences in facial expressions by domain of the age group of people [35]. Thus, starting with our study, various emotional change factors that can be

obtained from humans will be grafted into deep learning models for inter-ID variation analysis for FER. Refer to **Appendix D** for the expression tendency experiments according to demographics.

## 6 Conclusion and Others

To analyze the inter-identity variation of facial expressions, we newly propose an identity matching strategy based on optimization theory. The proposed method, which finds and utilizes optimal ID pairs based on similarity useful for facial expression learning, succeeded in explicitly dealing with identity shifts, that is, the main cause of performance gap in the test set. The results of verifying identity shifts and visualizing them from t-SNE perspective support the assumption of this paper that it is effective to analyze inter-identity variation by considering identity as a domain.

**Limitations.** The proposed method interacts with multiple target IDs based on a single reference ID. Thus, model learning may depend on selection of a reference ID. In order to overcome this limitation, dynamic connection methods of ID pairs along with more advanced relevance weight calculation methods should be studied in the future.

**Potential negative impacts.** FER may face with various privacy concerns. For example, someone can estimate and monitor an individual's emotional response in social media through this FER system without permission. Although only an authorized database with informed consent is used in this research field, someone's facial images are likely to be used for learning of the FER system without permission. In order to solve this problem, the privacy-aware face data processing and expression learning framework should be urgently studied and introduced.

## Acknowledgments and Disclosure of Funding

The authors greatly thank the three anonymous reviewers for their kind comments which have really helped improve the quality of our paper. This work was supported by IITP grants funded by the Korea government (MSIT) (No. 2021-0-02068, AI Innovation Hub and RS-2022-00155915, Artificial Intelligence Convergence Research Center (Inha University)), and was supported by the NRF grant funded by the Korea government (MSIT) (No. 2022R1A2C2010095 and No. 2022R1A4A1033549).

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
