# Optimal Transport-based Identity Matching
# for Identity-invariant Facial Expression Recognition
# (Appendix)

**Daeha Kim**
Inha University
kdhht5022@gmail.com

**Byung Cheol Song**
Inha University
bcsong@inha.ac.kr

The appendix describes the computation procedure of Sinkhorn-Knopp, the details of ELIM, the additional qualitative experimental results, and the pseudo-code, which were not covered by the main paper.

## 1 Details about Sinkhorn-Knopp

This section deals with the background and details of Sinkhorn-Knopp [2]. In general, optimal transport (OT) problem is regarded as a linear program (LP) and its solution is found through the interior point method. However, this interior point method, which involves factorization of KKT matrix, requires near-cubic complexity [1]. Thus, we modify the optimization method by adding an entropy constraint to the OT problem as follows:

$$X^* := \min_X \sum_{i,j} C_{i,j} X_{i,j} - \varepsilon \mathcal{H}(X),  \tag{1}$$

where $\varepsilon \in [0, +\infty]$ and $\mathcal{H}(X)$ is the entropy of $X$. Eq. 1 is called an entropy-regularized OT (EOT) problem, and its solution can be also found through Sinkhorn-Knopp [2] based on iterative updates of vectors. The following Lemma 1 guarantees the convergence of Eq. 1 and the uniqueness of $X$ [10].

**Lemma 1 [2].** *Let's define two non-negative vectors as $\boldsymbol{u}$ and $\boldsymbol{v}$, respectively. The flow matrix $X$ calculated through iterative updates of $\boldsymbol{u}$ and $\boldsymbol{v}$ is unique: $X = \mathrm{diag}(\boldsymbol{u})K\mathrm{diag}(\boldsymbol{v})$. Here, the scaled cost matrix $K$ is defined by the element-wise exponential of $-C/\eta$, and $\eta$ is the balancing factor. $\mathrm{diag}(\cdot)$ is a diagonalization operation.*

Eq. 1 is theoretically guaranteed to have a unique solution according to Lemma 1. $X$ of Eq. 1 is efficiently computed using Sinkhorn-Knopp's fixed point iteration as follows:

$$(\boldsymbol{u}, \boldsymbol{v}) \leftarrow (\boldsymbol{a}./K\boldsymbol{v}, \boldsymbol{b}./K^T\boldsymbol{u}),  \tag{2}$$

where $./$ stands for the element-wise division. Given $K$ and the relevance weights $\boldsymbol{a}$ and $\boldsymbol{b}$, the solution $X^*$ of Eq. 1 is obtained by iterating Eq. 2 enough times to converge. This process of updating $\boldsymbol{u}$ and $\boldsymbol{v}$ alternately can be simplified to a single step: $\boldsymbol{u} \leftarrow \boldsymbol{a}./K(\boldsymbol{b}./K^T\boldsymbol{u})$. Furthermore, when all elements of $\boldsymbol{a}$ are positive, this single step can be further simplified as follows [2]:

$$\boldsymbol{u} \leftarrow 1./(\tilde{K}(\boldsymbol{b}./K^T\boldsymbol{u})),  \tag{3}$$

where $\tilde{K} = \mathrm{diag}(1./\boldsymbol{a})K$. The overall flow of Sinkhorn-Knopp is described by Algorithm 1:

## 2 Details of Model Design

ELIM mainly consists of a backbone ($f_\phi$) and a projector ($f_\theta$). We considered three kinds of backbone network: AlexNet [7], ResNet18 [4], and Mlp-Mixer [11]. The backbone receives facial images,

---

**Algorithm 1** Entropy-regularized OT problem via Sinkhorn-Knopp

---

**Require:** Cost matrix $C$, weights $\boldsymbol{a}$, $\boldsymbol{b}$
**Require:** Balancing factor $\eta = 5$, counting value $t = 0$, max count $T = 5$
 1: Initialize $K = e^{-C/\eta}$, $\tilde{K} = \mathrm{diag}(1./\boldsymbol{a})K$, $\boldsymbol{u}_{old} = \mathbf{1}_n/n$
 2: **while** $\boldsymbol{u}$ changes or $t$ is less than $T$ **do**
 3: $\quad \boldsymbol{u} \leftarrow 1./(\tilde{K}(\boldsymbol{b}./K^T\boldsymbol{u}_{old}))$
 4: $\quad \boldsymbol{u}_{old} = \boldsymbol{u}$
 5: $\quad t \leftarrow t + 1$
 6: **end while**
 7: Get optimal value $\boldsymbol{u}^* = \boldsymbol{u}$
 8: $\boldsymbol{v}^* = \boldsymbol{b}./K^T\boldsymbol{u}^*$
 9: Output optimal flow matrix $X^* = \mathrm{diag}(\boldsymbol{u}^*)K\mathrm{diag}(\boldsymbol{v}^*)$

---

extracts feature maps, and encodes feature vectors through the so-called 'flatten' operation. Then, the projector designed using fully-connect (FC) layers with bias maps the feature vectors into a two-dimensional (2D) VA space. The parameter sizes of AlexNet, ResNet18, and Mlp-Mixer are 2.5M, 11M, and 59M, respectively. And the parameter size of the projector is about 88K.

# 3 Details of Metrics

RMSE, SAGR, PCC, and CCC were employed as performance metrics to evaluate ELIM in this paper. Given a ground-truth (GT) label set $Y$, a predicted label set $\hat{Y}$, and their corresponding mean and standard deviation, i.e., $(\rho_Y, \chi_Y)$ and $(\rho_{\hat{Y}}, \chi_{\hat{Y}})$, respectively, the metrics are defined as follows:

• Root mean-squared error (RMSE) measures the point-wise difference:

$$\mathrm{RMSE}(Y, \hat{Y}) = \sqrt{\mathbb{E}((Y - \hat{Y})^2)}.$$

• Sign agreement (SAGR) measures the overall positive and negative degree of emotion:

$$\mathrm{SAGR}(Y, \hat{Y}) = \frac{1}{n^{te}} \sum_{i=1}^{n^{te}} \Gamma\left(\mathrm{sign}\left(\boldsymbol{y}_i\right), \mathrm{sign}\left(\hat{\boldsymbol{y}}_i\right)\right),$$

where $n^{te}$ represents the number of samples in the test dataset. $\Gamma$ is an indicator function that outputs 1 if two values have the same sign, and 0 otherwise. In order to overcome the disadvantage of the previous metrics that cannot measure the correlation between two values, the following metrics were additionally adopted for evaluation purpose.

• Pearson correlation coefficient (PCC) measures the correlation between predictions and GT values:

$$\mathrm{PCC}(Y, \hat{Y}) = \frac{\mathbb{E}[(Y - \rho_Y)(\hat{Y} - \rho_{\hat{Y}})]}{\chi_Y \chi_{\hat{Y}}}.$$

• Concordance correlation coefficient (CCC) includes PCC and measures the agreement of two inputs by additionally considering the difference between their means:

$$\mathrm{CCC}(Y, \hat{Y}) = \frac{2\chi_Y \chi_{\hat{Y}} \mathrm{PCC}(Y, \hat{Y})}{\chi_Y^2 + \chi_{\hat{Y}}^2 + (\rho_Y - \rho_{\hat{Y}})^2}.$$

Next, the total regression loss for the training phase, i.e., $\mathcal{L}_{total}$ is defined as:

$$\mathcal{L}_{total}(Y, \hat{Y}) = \mathcal{L}_{main}(Y, \hat{Y}) + \mathcal{L}_{va}(Y, \hat{Y}) + 0.5\left(\mathcal{L}_{PCC}(Y, \hat{Y}) + \mathcal{L}_{CCC}(Y, \hat{Y})\right),$$

where $\mathcal{L}_{va} = \mathrm{MSE}_{val}(Y, \hat{Y}) + \mathrm{MSE}_{aro}(Y, \hat{Y})$, $\mathcal{L}_{PCC} = 1 - \frac{\mathrm{PCC}_{val}(Y, \hat{Y}) + \mathrm{PCC}_{aro}(Y, \hat{Y})}{2}$, and $\mathcal{L}_{CCC} = 1 - \frac{\mathrm{CCC}_{val}(Y, \hat{Y}) + \mathrm{CCC}_{aro}(Y, \hat{Y})}{2}$.

# 4 Additional Experimental Results

This section further verifies the identity-invariant FER capability of ELIM through experimental results that were not shown in the main paper. Note that as in the main paper, all experiments were performed using the AFEW-VA dataset and AL-tuned configurations.

## 4.1 Analysis of Inconsistency Cases

First, Figs. 1 and 2 provide additional experimental results for identity shift cases. In the case of neutral expression examples where GT is located close to the origin of VA space, not only ELIM but also CAF and baseline show consistent prediction results (see Fig. 1(b)). On the other hand, in an example of sad emotion, only ELIM accurately estimates GT (see Fig. 1(c)). Looking at the example of (strong) angry emotion in Fig. 1(d), only ELIM estimates GT relatively accurately. However, we can observe some failure cases where all methods including ELIM have failed (see Figs. 2(f) and 2(g)). This limitation occurs because all the methods including ELIM whose loss was designed based on static images cannot track the emotional change of previous frames inherently. Considering that most wild FER databases such as AFEW-VA consist of video clips, utilizing even temporal information may bring out further performance improvement.

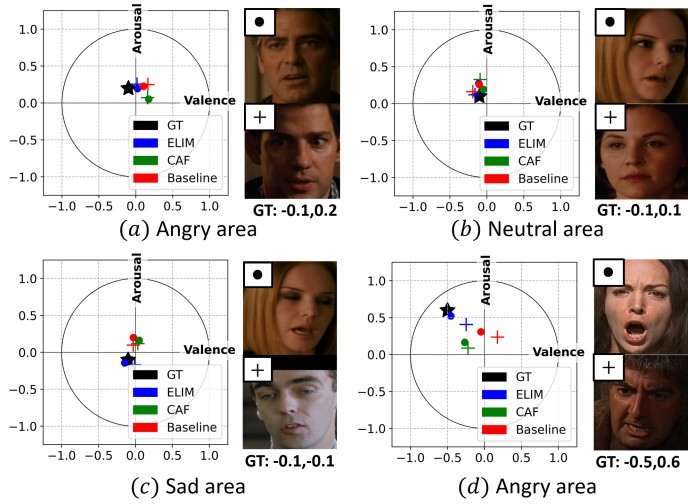

Figure 1: Results of identity shift when $p^\bullet(\boldsymbol{y}|\boldsymbol{z}) = p^+(\boldsymbol{y}|\boldsymbol{z})$ but $p^\bullet(\boldsymbol{z}) \neq p^+(\boldsymbol{z})$. Predictions of the two images are located in VA space as $\bullet$ and $+$, respectively. Best viewed in color.

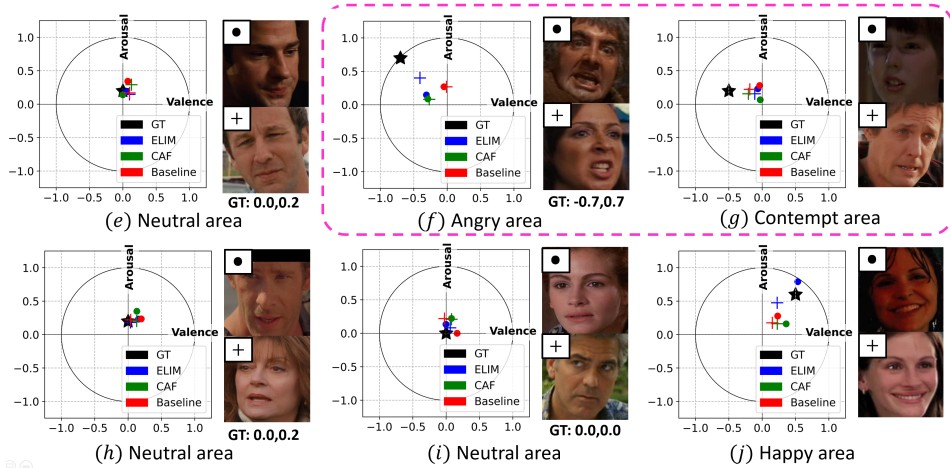

Figure 2: Additional results of identity shift. Dotted magenta box corresponds to failure cases.

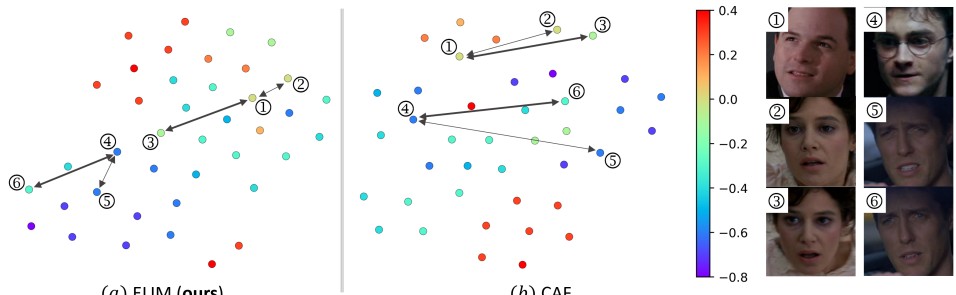

Figure 3: The learned feature distributions of ELIM and CAF. Samples were color-coded with valence values. Here, black lines indicate valence differences. The larger the difference, the thicker the line. Ten IDs randomly sampled from the test set were used for this experiment.

## 4.2 Visualization of ID-invariant Representation Ability

We visualized feature distributions of ELIM and CAF through t-SNE [12]. Fig. 3 compares the distance between a specific expression sample and two different expression samples with the same ID. In the case of ELIM, samples with different IDs but similar expressions are located closer in the feature space (see ①-② and ④-⑤ in Fig. 3(a)). Meanwhile, in CAF, samples with the same ID tend to be located close to each other (see ②-③ and ⑤-⑥ in Fig. 3(b)).

## 4.3 Comparing with Cross-attention Network

We implemented a cross-attention network that learns the similarity between ID samples from a part of the transformer encoder [13]. In detail, different ID feature sets of size $d \times n$ and $d \times m$, respectively, are passed through the multi-head attention (MHA) layer (self-attention). Then, the outputs are used together as inputs of another MHA layer (cross-attention). Next, the feature set of size $d \times m$ obtained through the cross-attention is passed through the FC layer with a one-dimensional output to generate an attention weight of size $m$. As a result, the ID shift of Eq. 3 of the main body is replaced by this weight. The experimental results for AFEW-VA are shown in Table 1.

This transformer encoder showed meaningful performance improvement in terms of RMSE (V) and CCC (V), which reflect the degree of positive and negative emotions, despite having small parameters of less than 100K. This results support the fact that the weights generated by the cross-attention mechanism are compatible with the inter-ID matching process. Even though somewhat weak performance was observed in the arousal axis, which reflects the degree of emotional activation, this attention method will be very useful for matching samples showing different emotional expressions between IDs.

Table 1: Validation experiment of cross-attention network on the AFEW-VA.

| Methods | RMSE (V) | CCC (V) | RMSE (A) | CCC (A) |
|---|---|---|---|---|
| ELIM-Sinkhorn | 0.186 | 0.602 | 0.198 | 0.581 |
| ELIM-Attention | 0.182 | 0.656 | 0.217 | 0.552 |

## 4.4 Expression Tendency according to Demographics

The proposed method assumes that the main cause of facial expression change is ID shift. On the other hand, depending on changes in age, gender, race, etc., the expression tendency may shift. In order to analyze the effect of shift caused by the demographics on FER performance, we newly designed **ELIM-Age**. ELIM-Age classifies samples by age group as domain instead of ID. Except for this process, ELIM-Age is the same as ELIM. The annotation and details of age labels were used as it is in [9]. Table 2 shows the experimental result of ELIM-Age on the AffectNet. Surprisingly, ELIM-Age showed better RMSE and CCC performance than all techniques including CAF. This results suggest the following two facts: 1) It is meaningful to deal with the shift in expression tendency

due to demographics such as age group as well as ID. 2) The optimal matching of ELIM is also effective in matching between age groups.

Table 2: Experimental results on the AffectNet.

| Methods | RMSE (V) | CCC (V) | RMSE (A) | CCC (A) |
|---|---|---|---|---|
| Baseline [8] | 0.37 | 0.60 | 0.41 | 0.34 |
| Kossaifi *et al.* [6] | 0.35 | 0.71 | 0.32 | 0.63 |
| Hasani *et al.* [3] | 0.267 | 0.74 | 0.248 | 0.85 |
| CAF (AL) [5] | 0.222 | 0.80 | 0.192 | 0.85 |
| CAF (R18) [5] | 0.219 | 0.83 | 0.187 | 0.84 |
| ELIM-Age (AL) | 0.221 | 0.821 | 0.187 | 0.856 |
| ELIM-Age (R18) | 0.209 | 0.835 | 0.174 | 0.866 |

## 5 Algorithms

The PyTorch-style pseudo-code for the overall learning process of ELIM is as follows.

---
**Algorithm 2** PyTorch-style pseudo-code for ID Grouping
---
```
# input: feat (b, d), label (b, 2)
# Note: mini-batch size (b), feature dimension (d)

IDfeat={}, IDlabel={} # Initialize dictionary
for i in range(N):
  indexes = ID-path(i)
  IDfeat[i], IDlabel[i] = feat[indexes], label[indexes]

return IDfeat, IDlabel
```
---

---
**Algorithm 3** PyTorch-style pseudo-code for Gumbel Sampling
---
```
# input: IDfeat {N,n,d}, IDlabel {N,n,2}
# Note: number of samples for each ID (n); Varies by ID.

for i in range(N):
  pi = norm(IDlabel[i], dim=1) # skip detailed process for convenience
  g = gumbel-softmax(pi.log())
  indexes = topk(g) # indexes is a list of k items (k≤n)
  IDfeat[i], IDlabel[i] = IDfeat[i][indexes], IDlabel[i][indexes]

return IDfeat, IDlabel
```
---

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

**Algorithm 4** PyTorch-style pseudo-code for ELIM

```
# input: image (b, c, h, w), label (b, 2)
# input: backbone (e.g., AlexNet), projector (FC layer)
# Hyperparameter: number of ID (N), metric (cosine distance)
# Loss function: Mean-squared error (MSE), PCC and CCC metrics (PCC-CCC)

feat = backbone(image)

# Note: IDfeat and IDlabel are in dictionary format for each ID.
IDfeat, IDlabel = ID-Grouping(feat,label)
IDfeat, IDlabel = Gumbel-sampling(IDfeat,IDlabel)

IDvec = Sinkhorn-Knopp(IDfeat,metric) # Fast iterative tool for OT

main-loss = 0
for i in range(N):
  if i is target ID index:
    IDfeatnorm = ID-normalization(IDfeat[i],IDvec[i])
    IDpredictions = projector(IDfeatnorm)
    main-loss += MSE(IDpredictions, IDlabel[i]) # Risk minimization loss

predictions = projector(feat)
va-loss = MSE(predictions, label) # conventional MSE loss
pcc-ccc-loss = PCC-CCC(predictions, label)

total-loss = main-loss + va-loss + pcc-ccc-loss
total-loss.backward # update model parameters
```

[6] Jean Kossaifi, Antoine Toisoul, Adrian Bulat, Yannis Panagakis, Timothy M Hospedales, and Maja Pantic. Factorized higher-order cnns with an application to spatio-temporal emotion estimation. In *Proceedings of the IEEE/CVF Conference on Computer Vision and Pattern Recognition*, pages 6060–6069, 2020.

[7] Alex Krizhevsky, Ilya Sutskever, and Geoffrey E Hinton. Imagenet classification with deep convolutional neural networks. In *Advances in neural information processing systems*, pages 1097–1105, 2012.

[8] Ali Mollahosseini, Behzad Hasani, and Mohammad H Mahoor. Affectnet: A database for facial expression, valence, and arousal computing in the wild. *IEEE Transactions on Affective Computing*, 10(1):18–31, 2017.

[9] Rafael Poyiadzi, Jie Shen, Stavros Petridis, Yujiang Wang, and Maja Pantic. Domain generalisation for apparent emotional facial expression recognition across age-groups. *arXiv preprint arXiv:2110.09168*, 2021.

[10] Richard Sinkhorn. Diagonal equivalence to matrices with prescribed row and column sums. ii. *Proceedings of the American Mathematical Society*, 45(2):195–198, 1974.

[11] Ilya O Tolstikhin, Neil Houlsby, Alexander Kolesnikov, Lucas Beyer, Xiaohua Zhai, Thomas Unterthiner, Jessica Yung, Andreas Steiner, Daniel Keysers, Jakob Uszkoreit, et al. Mlp-mixer: An all-mlp architecture for vision. *Advances in Neural Information Processing Systems*, 34, 2021.

[12] Laurens Van der Maaten and Geoffrey Hinton. Visualizing data using t-sne. *Journal of machine learning research*, 9(11), 2008.

[13] Ashish Vaswani, Noam Shazeer, Niki Parmar, Jakob Uszkoreit, Llion Jones, Aidan N Gomez, Ł ukasz Kaiser, and Illia Polosukhin. Attention is all you need. In *Advances in Neural Information Processing Systems*, volume 30, pages 5998–6008. Curran Associates, Inc., 2017.