# OpenReview forum: "Optimal Transport-based Identity Matching for Identity-invariant Facial Expression Recognition"
_NeurIPS.cc/2022/Conference — NeurIPS 2022 Accept_

### Official Review · Reviewer_zNEr · 2022-07-07

**Rating:** 6
**Confidence:** 4
**Soundness:** 3 good
**Presentation:** 3 good
**Contribution:** 3 good

**Summary:**

The paper proposes an approach to identity invariant facial expression recognition. Identity matching is formulated as an optimal transport problem. The optimal flow for this problem is found using Sinkhorn-Knopp. Evaluations are conducted on three public FER datasets.

**Questions:**

How can the proposed approach handle shift due to demographics such as age, gender, or race?

**Limitations:**

Negative impact has been addressed. Limitation regarding the reference ID is detailed, however, limitations on the scope of demographics and the proposed approach are not detailed.

**Strengths And Weaknesses:**

Strengths

Paper proposes a solution to a challenging problem in FER. Problem is well formulated in paper including needed background information to setup proposed solution. State of the art results are detailed on multiple public datasets. Results show the proposed approach can easily be integrated into different models. Ablation study on the relevance weights, and impact of number of IDs is well conducted.

Weaknesses

As the paper focuses on identity invariant FER, other factors such as age, gender, and race could have a large impact on the ID matching and FER learning. While this is briefly discussed in Section 5.4 (A5), the discussion is largely superficial with a tech report being cited that talk about differences in facial expression across age. It is not clear how the proposed approach can handle these changes across different demographics. Handling this type of shift is important for generalized FER.

---

> ### Author Response · Authors · 2022-08-01
> **Answer to question from reviewer zNEr**
>
> The proposed method assumes that the main cause of facial expression change is ID shift. On the other hand, depending on changes in age, gender, race, etc., the expression tendency may shift (cf. Line 299-307). In order to analyze the effect of shift caused by the demographics on FER performance, we newly designed ELIM-Age. ELIM-Age classifies samples by age group as domain instead of ID. Except for this process, ELIM-Age is the same as ELIM. The annotation and details of age labels were used as it is in [1]. The experimental result of ELIM-Age for AffectNet is as follows. Surprisingly, ELIM-Age showed better RMSE and CCC performance than all techniques including CAF. This result suggests the following two facts: 1) It is meaningful to deal with the shift in expression tendency due to demographics such as age group as well as ID. 2) The optimal matching of ELIM is also effective in matching between age groups.
>
> | Methods        | RMSE-V | CCC-V | RMSE-A | CCC-A |
> | :---------------------: | :---------: | :-------: | :---------: | :------: |
> | Baseline [2]     | 0.37    | 0.60   | 0.41    | 0.34  |
> | Kossaifi et al. [3] | 0.35    | 0.71   | 0.32    | 0.63  |
> | Hasani et al. [4] | 0.267   | 0.74   | 0.248   | 0.85  |
> | CAF (AL) [5]    | 0.222   | 0.80   | 0.192   | 0.85  |
> | CAF (R18) [5]   | 0.219   | 0.83   | 0.187   | 0.84  |
> | ELIM-Age (AL)  | 0.221   | 0.821  | 0.187   | 0.856 |
> | ELIM-Age (R18) | 0.209   | 0. 835  | 0.174   | 0.866 |
>
> Thanks to the reviewer's good comments, we were able to design experiments that could further enhance the value of this paper. If this paper is accepted, the source code as well as demos and experimental results will be uploaded immediately.
>
> References
>
> [1] R. Poyiadzi et al., Domain generalization for apparent emotional facial expression recognition across age-groups, ArXiv 2021.
>
> [2] A. Mollahosseini et al., Affectnet: A database for facial expression, valence, and arousal computing in the wild, TAC 2017.
>
> [3] J. Kossaifi et al., Factorized higher-order cnns with an application to spatio-temporal emotion estimation, CVPR 2020.
>
> [4] B. Hasani et al., Breg-next: Facial affect computing using adaptive residual networks with bounded gradient. TAC 2020.
>
> [5] D. Kim and BC Song, Contrastive adversarial learning for person independent facial emotion recognition. AAAI 2021.

---

> > ### Comment · Reviewer_zNEr · 2022-08-08
> > **Update experiments**
> >
> > Updated experiments for age look good. This is an interesting paper, that I think can be accepted.

---

> > > ### Author Response · Authors · 2022-08-08
> > > **Response to reviewer zNEr's reply**
> > >
> > > Thanks for the reviewer's positive response.
> > >
> > > If this paper is accepted, the software link including this experiment and demo will be added to the camera-ready version.

---

### Official Review · Reviewer_U2SY · 2022-07-08

**Rating:** 6
**Confidence:** 5
**Soundness:** 3 good
**Presentation:** 2 fair
**Contribution:** 3 good

**Summary:**

The paper proposes a method for facial expression recognition (by means of valence and arousal) which aims to remove the factors that relate to identity from the backbone features, so as to make the classifier be robust against identity-specific features. In particular, a method is proposed to remove from the feature representation the identity shift, by computing the optimal transport between the batch-specific features in a way to represent the image-specific domain shift from the other images in the batch. The features are then normalized according to the computing shifting mean and scale and passed through the final classifier which regresses the values of valence and arousal. The use of optimal transport to assign the image-specific mean vector is grounded on the idea that the batch-specific features can be rearranged, while preserving the total mass distribution, so as to remove the identity-based similarities encountered in the batch. The method is validated in a set of challenging datasets achieving competitive results.

**Questions:**

As stated above, my main concerns or questions can be summarized as:
1) Exploring or comparing against of attention-based methods for residual learning.

2) Inference

3) Use of SEWA or AffectNet to compare against CAF

4) Different number of parameters for same networks in Table 1.

5) Different losses and comparison against existing methods.

6) Explanation of Figure 4


**Limitations:**

The limitations are properly addressed in the paper and hence I have no further questions in this regard.

**Strengths And Weaknesses:**

The paper is technically novel and achieves competitive results and hence has the merits to be accepted at NeurIPS. The use of OT to disentangle the factors of identity is novel, the motivation is clear and the experiments are compelling. The paper is well accompanied by algorithms and descriptions that help understanding the derivation, with further pseudo-code for the crucial parts in the Supplementary Material. The paper could benefit from a better clarity of presentation and elaboration on the significance and meaning of the learned shift, and thus I would like to particularly ask the authors to improve the manuscript’s presentation and writing substantially for better clarity.

My main comments and concerns are listed below:

- Digging under the hood of Equations 3 and 4 as well as how the cost matrix is computed based on similarity, I can spot that there are some substantial similarities and differences between using the OT assignment and using cross-attention to attenuate the influence of the global features. Plugging the cost matrix into Eqn 3 and observing the “residual” behaviour of Eqn. 4 can derive to expressing the new features as a linear combination of the other images’ features, given by a weighted similarity between them. This resembles to me the use of cross-attention with X being the attention weights. Obviously, the authors propose a completely different way to assign these attention weights, but after reading the full derivation I would like to ask the authors if a small transformer encoder instead to compute the mean features in Equation (3) would also enforce the learned features to be identity-invariant. I think that such exploration is important and the paper would gain significant strength such comparison would be given.

- It is not mentioned in the paper (or I missed that), but there is no reference to the process of inference. My understanding is that at test time no shift is obtained because the features are already expected to be “centered”, but some clarification in this regard would be appreciated.

- I think the paper would gain strength by including the performance of the proposed approach on AffectNet and SEWA, while AffWild2 and AffWild are challenging, they are a subset of each other, and AFEW-VA has recently given way to the aforementioned datasets.

- What makes the number of parameters on Table 1 be different for the proposed method and that of CAF for the AL-tuned and ResNet-18 backbones?

- I am a bit confused about the results shown in Figure 5. First, the behaviour of the curves is a bit counterintuitive as e.g. there is a drop in two points of CCC between using 18 and 19 subjects, which is recovered after using 20 subjects. What is this behaviour due to? In addition, it is not clear to me what does w/o L_va mean, considering that this is the main training loss. The blue curve, does it correspond to a training using the PCC + CCC losses? Please do clarify.

- It is important also to understand the contribution of the training losses in the method. How does this compare against competing methods? If they had been trained without L_ccc or P_ccc then I believe the authors should also consider a fair evaluation using their method trained with the same losses. This will indicate the contribution of the losses in the performance gain.

- I didn’t get quite well what Figure 4 attempts to represent, so I would like to request the authors to elaborate on it.

- Finally, as mentioned before, I believe the paper would greatly benefit from better writing and presentation. The readability of the paper is somewhat poor affecting its reach.

---

> ### Author Response · Authors · 2022-08-01
> **Responses to reviewer U2SY's weakness points**
>
> 1. Many thanks for the helpful comments from the reviewer. Comparative analysis between Sinkhorn, which does not require trainable parameters, and cross-attention using parameters, may highlight the value of the proposed method. Thus, we implemented a cross-attention network that learns the similarity between ID samples from a part of the transformer encoder [1]. In detail, different ID feature sets with sizes of $d\times n$ and $d\times m$, respectively, are passed through the multi-head attention (MHA) layer (self-attention). Then, the outputs are used together as inputs of another MHA layer (cross-attention). Here, notation is the same as those of Line 141 of the main body. Next, the feature set of $d\times m$ size obtained through cross-attention is passed through the FC layer with a one-dimensional output to generate an attention weight of the $m$ size. As a result, the ID shift in Eq. 3 is replaced by this weight. The experimental result for AFEW-VA is shown in the table below.
>
> | Methods | RMSE-V | CCC-V | RMSE-A | CCC-A |
> | :--- | :---: | :---: | :---: | :---: |
> | ELIM (AL) | 0.186 | 0.602 | 0.198 | 0.581 |
> | ELIM (AL)-Att. | 0.182 | 0.656 | 0.217 | 0.552 |
>
> - This transformer encoder showed meaningful improvement in terms of RMSE-V and CCC-V, despite small parameters of less than 100K. This supports the fact that the weights generated by the cross-attention mechanism are compatible with the inter-ID matching process. Even though somewhat weak performance was observed in the arousal axis, which reflects the degree of emotional activation, this attention method will be very useful for matching samples showing different emotional expressions between IDs. This experimental result has been included in the revised Appendix.
>
> 2. In the last line of the caption of Figure 2, the contents of ELIM inference are already mentioned. For easy understanding, we mentioned additional details of inference on Line 216 of the revised manuscript.
>
> 3. Unlike AffWild(/2), which consists of several samples per ID, all facial images of AffectNet have different IDs. So, we newly designed “ELIM-Age” that groups samples by age group as domain instead of ID. Except for this preprocessing, ELIM-Age and ELIM are equivalent. The annotation and details of age labels were used as it is in [2].
>
> - The experimental result below implies the following two facts: 1) The learning mechanism of ELIM is valid even in AffectNet. 2) Dealing with shifts due to demographics such as age and ID should be also considered important in FER field.
>
> | Methods|RMSE-V|CCC-V|RMSE-A|CCC-A |
> | :--- | :---: | :---: | :---: | :---: |
> | Baseline [3]|0.37|0.60|0.41|0.34 |
> | Kossaifi et al. [4]|0.35|0.71|0.32|0.63 |
> | Hasani et al. [5]|0.267|0.74|0.248|0.85 |
> | CAF (AL) [6]|0.222|0.80|0.192|0.85 |
> | CAF (R18) [6]|0.219|0.83|0.187|0.84 |
> | ELIM-Age (AL)|0.221|0.821|0.187|0.856 |
> | ELIM-Age (R18)|0.209|0. 835|0.174|0.866 |
>
> - As for SEWA, we will test it as soon as it is approved for use and show the final result.
>
> 4. The number of backbone parameters is the same in both CAF and ELIM. However, CAF utilizes a discriminator network, whereas ELIM utilizes a projector ($f_\theta$ in Fig. 2), which is slightly larger than CAF. This may cause a difference in the total number of parameters of ELIM and CAF.
>
> 5. The cause of the performance variation is randomness due to the statistically selected mini-batch samples and reference IDs. w/o $L_{va}$ means that $L_{va}$ of Line 169 is excluded from model training. Both experiments in Fig. 5 include $L_{pcc}$ and $L_{ccc}$. We partly described this fact in the caption of Figure 5 of the revised manuscript.
>
> 6. Two previous works, i.e., HO-Conv and CAF that compete with the proposed method for SOTA performance basically use $L_{pcc/ccc}$. However, the other techniques such as [7] do not employ $L_{pcc/ccc}$. So, for fair comparison, we performed an ablation study on $L_{pcc/ccc}$. The experimental result for AFEW-VA is as follows. In the absence of $L_{pcc/ccc}$, the CCC performance decreased by about 0.15 in both the Valence and Arousal axes. Note that despite this performance degradation, the proposed method still outperforms [7] in all indicators.
>
> | Methods | PCC-V | PCC-A | CCC-V | CCC-A |
> | :--- |:---: |:---: |:---: |:---: |
> | Mitenkova et al. [7] | 0.33 | 0.42 | 0.33 | 0.40 |
> | ELIM w/ $\mathcal{L}_{pcc/ccc}$ | 0.680 | 0.645 | 0.602 | 0.581 |
> | ELIM w/o $\mathcal{L}_{pcc/ccc}$ | 0.620 | 0.532 | 0.453 | 0.429 |
>
> 7. According to the reviewer’s comment, we modified the caption of Figure 4 of the revised manuscript and some parts of Lines 236-238.
>
> References
>
> [1] A Vaswani et al., NeurIPS 2017.
>
> [2] R. Poyiadzi et al., ArXiv 2021.
>
> [3] A. Mollahosseini et al., TAC 2017.
>
> [4] J. Kossaifi et al., CVPR 2020.
>
> [5] B. Hasani et al., TAC 2020.
>
> [6] D. Kim and BC Song, AAAI 2021.
>
> [7] A. Mitenkova et al., FG 2019.

---

> > ### Comment · Reviewer_U2SY · 2022-08-08
> > **Response to review**
> >
> > I have carefully read the comments to my review and concerns, and I am overall happy with the authors' response. I acknowledge and appreciate the effort towards carrying over the mentioned experiments which as the results indicate show some extra insight into the proposed method.
> >
> > I can observe that attention methods would be a valid alternative method to that proposed despite the increased cost, and as such this option opens an interesting research topic.
> >
> > The dependency of the proposed method with the pcc loss is quite impressive as the results seem to be crucially affected by removing it. Does this also apply to the competing methods beyond that of [7]?
> >
> > Provided the response I still believe the paper is solid enough to be considered for publication and hence I see no reason to change my score.

---

> > > ### Author Response · Authors · 2022-08-09
> > > **Response to reviewer U2SY's reply**
> > >
> > > Thank you for the reviewer's positive response. In the future, we will be able to design a robust parametric model for inter-ID matching derived from attention method(s), and this direction must be a valuable research topic. Also, thanks for the nice question. For example, $L_{ccc}$ has been often used in FER studies (e.g., workshop challenge) aimed at improving performance [1]. In other words, $L_{ccc}$ (or $L_{pcc}$) already plays an important role as a regularization loss function for performance improvement in the FER field. If this paper is accepted, a software link including additional experiments and demos will be added to the camera-ready version.
> > >
> > > Reference
> > >
> > > [1] V. Karas et al., Time-Continuous Audiovisual Fusion with Recurrence vs Attention for In-The-Wild Affect Recognition, CVPRW 2022.

---

### Official Review · Reviewer_yjnn · 2022-07-11

**Rating:** 4
**Confidence:** 4
**Soundness:** 2 fair
**Presentation:** 2 fair
**Contribution:** 2 fair

**Summary:**

The paper proposes to quantify the inter-identify variation by utilizing pairs of similar expressions for identify-invariant facial expression recognition. To find pairs of similar expressions from different identities, the authors define the inter-feature similarity as a transportation cost. Then, they perform optimal identity matching to find the optimal flow with minimum transportation cost by Sinkhorn-Knopp iteration.

**Questions:**

1. Please give reasonable explanations for the above weaknesses.
2. In line 169, I do not find the definition of $L_{va}$ in the cited paper [50]. Maybe the author made a mistake.

**Limitations:**

Please see the Questions.

**Strengths And Weaknesses:**

Strengths:
1. The authors propose a method to quantify the inter-identify variation, which is different from the previous works in a non-quantitative way.
2. The experimental results exceed most existing works
Weaknesses:
1. As mentioned in the paper, inter-identify variation is quantified by pairs of similar expressions. However, when the expressions of individuals with different IDs differ greatly in a batch, that is, the precondition fails. The methods may not work.
2. The variations between reference and targets are calculated in a batch, but different references will be chosen in different batches. The variations among references are not considered.
3. Although the authors describe the training process in detail, they do not mention the testing process, which I believe is a non-trivial process.
4. The authors only verify that the proposed method performs better in VA prediction, but it is lack experiments on expression and AU prediction.

---

> ### Author Response · Authors · 2022-08-01
> **Responses to reviewer yjnn's weakness points**
>
> 1. Many thanks for the good comment from the reviewer. Considering such a situation, we adopted a solution in terms of configuration. That is, when training our model, we allocated a sufficiently large mini-batch size (cf. Line 207-208). As a result, we did not find an extreme case where the facial expressions of all samples between IDs are totally different each other. If the valence signs of facial expressions between ID groups are all different, ID-dependent features can be obtained by Eq. 4 due to the matching between samples of different facial expressions. However, as mentioned above, this phenomenon will happen extremely rarely. In addition, since ELIM finds the model solution through iterative training, such a case is hard to affect the overall performance. When training our model, we examined the validity of matched samples between IDs in detail. The additional analysis and actual matching results were added to the revised Appendix.
>
> 2. In the limitations section (Section 6), we mentioned that the model training may be partially dependent on the selection of reference IDs (cf. Line 315-318). In order to explicitly analyze the effect of reference-target pair variation on overall performance, we changed the existing configuration of one-to-$N$ (ON) pair to one-to-one (OO) pair. The test results for the AFEW-VA dataset are as follows: We could observe the overall improvement of RMSE and CCC performance. Especially, there was an improvement of about 4% in terms of CCC-A. This proves that OO pairs composed of independent reference-target pairs are more advantageous for learning ID-invariant features than ON pairs.
>
> | Methods      | RMSE-V | CCC-V | RMSE-A | CCC-A |
> | :-------------------- | :--------: | :-------: | :--------: | :-------: |
> | ELIM (AL)-ON  | 0.186  | 0.602  | 0.198  | 0.581  |
> | ELIM (AL)-OO  | 0.190  | 0.610  | 0.192  | 0.622  |
>
> 3. In the last line of the caption of Figure 2, the details of inference are already mentioned. In detail, the prediction values are output through the backbone ($f_\phi$) and the affine projector ($f_\theta$). Note that at this time, feature normalization (Eq. 4) through ID shift is not used. For easy understanding, we added inference details on Line 216 of the revised manuscript.
>
> 4. The domain gap between VA FER and AU prediction is much larger than that between VA FER and category (emotion) label-based FER. Thus, it is practically impossible to apply the proposed method to AU prediction in this rebuttal. Instead, we designed an additional experiment on AffectNet in which both category and VA labels were annotated. Unfortunately, however, since the IDs of all facial images of AffectNet are different, AffectNet cannot be directly applied to ELIM where ID grouping is performed as a preprocessing. On the other hand, based on an analysis that the expression tendency can be shifted according to changes in age, gender, and race (cf. Line 299-307), we newly implemented the so-called ELIM-Age, which groups samples by age group as a domain. Except for a specific pre-processing process, ELIM-Age and ELIM are equivalent. Referring to [1], age-annotated AffectNet was applied to ELIM-Age.
>
> - We compared the proposed method with some existing methods that used both category and VA labels for evaluation. The proposed method showed better performance in all aspects such as RMSE, CCC, and class accuracy (%) than other methods. Although the proposed method showed somewhat lower accuracy than Face2Exp [5], which is the latest method targeting only category labels, this difference will be sufficiently overcome if further tuning for category labels is performed with sufficient time. Also, please note that this paper is an FER study focusing on dimensional models of emotion, i.e., VA space.
>
> | Methods | Acc. | RMSE-V | CCC-V | RMSE-A | CCC-A |
> | :--- | :---: | :---: | :---: | :---: | :---: |
> | Baseline [2] | 0.58 | 0.37 | 0.60 | 0.41 | 0.34 |
> | VGG-Face [3] | 0.60 | 0.37 | 0.62 | 0.39 | 0.54 |
> | HO-Conv [4] | 0.59 | 0.35 | 0.71 | 0.32 | 0.63 |
> | ELIM-Age (R18) | 0.611 | 0.209 | 0. 835 | 0.174 | 0.866  |
> | Face2Exp [5] | 0.64  | - | - | - | - |
>
> 5. The average of the MSE values between the predictions of the features that have not been normalized in Eq. 4 and the VA labels corresponds to $\mathcal{L}_{va}$. We described the details in Lines 169-170 of the revised manuscript.
>
> References
>
> [1] R. Poyiadzi et al., Domain generalization for apparent emotional facial expression recognition across age-groups, ArXiv 2021.
>
> [2] A. Mollahosseini et al., Affectnet: A database for facial expression, valence, and arousal computing in the wild, TAC 2017.
>
> [3] D. Kollias et al., Generating faces for affect analysis, ArXiv 2018.
>
> [4] J. Kossaifi et al., Factorized higher-order cnns with an application to spatio-temporal emotion estimation, CVPR 2020.
>
> [5] D. Zeng et al., Face2Exp: combating data biases for facial expression recognition, CVPR 2022.

---

> > ### Comment · Reviewer_yjnn · 2022-08-08
> > **Problems during inference**
> >
> > In training phase, $f_{(\varphi)}$ and $f_{(\theta)}$ are expected to better learn the identity and expression joint features, and the identity-invariant expression features from the normalized features, respectively. As far as I'm concerned, it is impossible that the model has the ability to directly obtain the identity-invariant expression features from the identify and expression joint features during inference.

---

> > > ### Author Response · Authors · 2022-08-09
> > > **Response to reviewer yjnn's reply**
> > >
> > > Many thanks for the good comments from the reviewer. Although feature normalization is not applied in the inference phase, we can expect positive performance. The reason is as follows:
> > > - The mechanism of allowing the model to learn domain-invariant features through training domains (i.e., training IDs in our case) and then performing naïve prediction with **the model trained** by the inference phase is the general setting of domain generalization studies such as [1]. The reason why this setting is valid is based on the assumption that a model that has completed training for sufficient domains will be robust even to unseen domains.
> > > - Also, since the domains in the inference phase have not been experienced during training phase, we can expect them to be already “in the center” without loss of generality.
> > >
> > > If this paper is accepted, this discussion will be added to the camera-ready version.
> > >
> > > Reference
> > >
> > > [1] K. Zhou et al., Domain Generalization with MixStyle, In ICLR 2021.

---

### Meta-Review · Area_Chair_Fqii · 2022-09-04

**Recommendation:** Accept
**Confidence:** Certain

**Metareview:**

Authors propose a new strategy for a hard problem that reviewers found compelling and novel. The  experimental details are complex and we encourage the authors to address the many issues the reviewers raise.

**Award:**

No

---

### Decision · Program_Chairs · 2022-09-14

Accept